# Preparation of a PVA/Chitosan/Glass Fiber Composite Membrane and the Performance in CO_2_ Separation

**DOI:** 10.3390/membranes13010036

**Published:** 2022-12-28

**Authors:** Yunwu Yu, Chunyang Xie, Yan Wu, Peng Liu, Ye Wan, Xiaowei Sun, Lihua Wang, Yinan Zhang

**Affiliations:** 1School of Materials Science and Engineering, Shenyang Jianzhu University, Shenyang 110168, China; 2Research Center for Nanotechnology, Changchun University of Science and Technology, Changchun 130022, China

**Keywords:** carbon dioxide, polyvinyl alcohol, chitosan, glass fiber microporous membrane, gas separation

## Abstract

In this study, a novel composite membrane was developed by casting the mixed aqueous solution of chitosan (CS) and polyvinyl alcohol (PVA) on a glass fiber microporous membrane. The polymeric coating of a composite membrane containing amino groups and hydroxyl groups has a favorable CO_2_ affinity and provides an enhanced CO_2_ transport mechanism, thereby improving the permeance and selectivity of CO_2_. A series of tests for the composite membranes were taken to characterize the chemical structure, morphology, strength, and gas separation properties. ATR-FTIR spectra showed that the chemical structure and functional group of the polymer coating had no obvious change after the heat treatment under 180 °C, while SEM results showed that the composite membranes had a dense surface. The gas permeance and selectivity of the composite membrane were tested using single gases. The results showed that the addition of chitosan can increase the CO_2_ permeance which could reach 233 GPU. After a wetting treatment, the CO_2_ permeance (454 GPU) and gas selectivity (17.71) were higher than that of dry membranes because moisture promotes the composite membrane transmission. After a heat treatment, the permeance of N_2_ decreased more significantly than that of CO_2_, which led to an increase in CO_2_/N_2_ selectivity (10.0).

## 1. Introduction

Over the past decade, membrane-based CO_2_ separation from a mixture with light gases, such as O_2_, N_2_ and CH_4_, has been a widely valued technology with industrial and technological advantages. The membrane-based separation technology is environmentally sustainable, has a simple process, has scale adaptability and a high energy efficiency. The membrane technology seems especially effective for CO_2_ separation and capture from a combustion flue gas [1,2,3]. In 2019 alone, carbon emissions reached approximately 1616 million metric tonnes due to the consumption of fossil fuels which accounted for about 80% of the energy production in the United States alone [1]. In response to the pressures of international carbon emissions reduction, carbon dioxide separation and capture represents the next opportunity for the deployment and practical application of gas separation membranes.

Many different kinds of materials have been applied in the preparation of gas separation membranes, including carbon molecular sieves, polymers, ceramics, clays and composite membranes [4,5,6]. Polymeric membranes have commercial application potential due to their flexibility, good processability and relatively low costs. The membranes made of glassy polymers with a high selectivity generally are preferred to rubbery polymer membranes [7]; however, the permeability of glassy polymer membranes is generally not high enough, especially the commercialized polymers, and they are also troubled by performance deterioration during the separation operation caused by the densification or plasticization of polymers [7,8]. Improving the permeability and selectivity with a trade-off relationship simultaneously by designing the chemical composition and structure of membranes is an efficient strategy to exceed Robson’s upper line.

For the above purposes of improving the separation performance, the preparation of composite membranes with a dense layer on the top of a porous substrate has become a potential option to improve the gas transmission, selectivity and mechanical properties [9]. The gas permeability of membranes is inversely proportional to the effective thickness of a dense selective layer, thus reducing the effective thickness of the selective layer to obtain practical gas separation membranes. On the other hand, the dense selective layer could provide a high selectivity, and then break the trade-off relationship between the permeability and selectivity [10]. Compared with conventional asymmetric membranes with an integrated dense skin, the composite membranes with a dense-layer and porous substrate have several outstanding advantages, for instance, much smaller amounts of materials are required to construct the selective layer, coating composition technology can optimize the membrane separation performance and morphology in each layer independently according to the performance and process requirements. Conversely, there are fewer requirements for the mechanical properties and processability of the materials forming a dense layer.

The gas separation properties of polymeric membranes have been investigated such as cellulose acetate [11], polysulfone [12], modified PEEK [13], and polyimides [14]. The homogeneous membranes based on these polymers suffer from a truncated CO_2_/N_2_ separation factor, have permeability, and some of these lack thermal stability. For example, as shown by [13], the CO_2_ permeability of PSF/PES blending membranes was generally lower than 30 GPU, at a 2 bar and 25 °C, while a cellulose acetate membrane with poor thermal stability showed a glass transition at about 185 °C which limited the treatment and use at high temperatures [11]. High gas permeability reduces the operating cost of membrane separation as it requires less areas of membranes to separate the same amount of gas mixture while a high selectivity increases the purity of the permeate gas. The membranes following a facilitated transport mechanism are preferred due to a high CO_2_ transmission and selectivity such as those membranes containing an hydroxyl group and amino group [15,16]. Polyvinyl alcohol (PVA) is a widely used biodegradable polymer with nontoxicity and a high hydrophilicity [17]. The existence of hydroxyl groups and interchain hydrogen bonds leads to a semi-crystalline aggregated structure in PVA with a low fractional free volume and permeability. In the application of a membrane preparation, the water-soluble PVA could be blended or cross-linked with plasticizers or other polymers resulting in the destruction of the semi-crystal structure and an improvement of the separation performance. Different reagents could be used as plasticizers or cross-linkers such as polyhydric alcohol, low-molecular-weight PEG, PEI citric acid, urea/ethanolammine and glutaraldehyde. A series of polyethylene glycol and glutamic acid/polyvinyl alcohol composite membranes with fixed hydroxyl and amino carriers was reported by Shiue et al. [18], with the best selection for CO_2_ separation being under the condition of a 1 bar pressure difference and a humid environment, where the selectivity reached 10.05. Ho et al. [19,20] published a series of research papers on CO_2_ separation via modified PVA membranes containing polyallylamine (PAAm), amino acid salt and KOH, which worked as mobile or fixed carriers. Moreover, a PAAm/PVA blending membrane reported by Cai et al. showed excellent CO_2_/N_2_ and CO_2_/CH_4_ selectivities which reached 80 and 58 at a 0.1 MPa feed gas pressure, respectively [21]. Additionally, a composite membrane containing a defect-free PVAm/PVA blend layer with a facilitated transport effect on a porous PSf substrate was prepared and evaluated in previous literature [22]. The ultra-thin PVAm/PVA layer was prepared via commercial PVAm and PVA (1788), and the CO_2_/N_2_ mixed-gas separation factor and CO_2_ permeance up to 174 and 0.58 m^3^(STP)/(m^2^·h·bar) were measured. It was suggested that the CO_2_ was transported according to the facilitated transport mechanism, while the PVAm/PVA blend composite membranes simultaneously exhibited outstanding stability during the 400 h separation performance test.

The biopolymer chitosan (CS) derived from the deacetylation of chitin, a polysaccharide very abundant in nature, is bio-renewable, chemically stable, antibacterial and hydrophilic. In addition, it is soluble in an acidic aqueous solution, biocompatible, biodegradable and non-toxic [23,24]. In the application of gas separation, chitosan has attracted increasing attention in recent years, due to the structure possessing a large proportion of hydroxyl groups and amino groups which could greatly improve the absorption of CO_2_ in a membrane [25,26]. The CO_2_ transportation in chitosan-based membranes occurs via the hopping mode caused by amine groups following the facilitated transport and solution-diffusion mechanism, in the meantime, N_2_ follows only the solution-diffusion mechanism [26]. The single gas permeation of CO_2_ and N_2_ in CS membranes crosslinked with trimesoyl chloride was reported where the CO_2_ permeability reached 163 Barrers with an ideal separation factor of 42 [27].

A glass fiber (GF) membrane shows many notable properties including a chemical inertness towards salts and most acids, has a low cost, a high mechanical and tensile strength, a high dimensional stability, corrosion resistance, and excellent thermal stability compared to polymeric porous membranes [28]. Moreover, the reactive silanol groups on glass fibers could react with various organosilane compounds and then introduce desired properties on glass surfaces easily [29], for instance, superhydrophobicity, superhydrophilicity, compatibility and so on. The above organosilane-modification method has been widely used in glass fiber (GF) membrane-based oil–water separation [30]. So far as we know, glass fiber porous membranes are rarely used for the preparation of composite gas separation membranes currently.

The purpose of this work was to prepare a technically viable polymer/glass fiber composite membrane containing a polymeric selective layer for CO_2_ separation. The advantages of this kind of composite membrane include having a good separation performance, fabrication process simplicity, and good mechanical strength. In the fabrication of defect-free composite membranes, there are some challenges that need attention, and one of them is the effect of substrate resistance. The property and morphology of a substrate is usually as important as the optimization of the defect-free selective layer and beside a strong mechanical and chemical stability, an ideal substrate should possess a highly porous membrane structure. In this study, the ultra-fine glass fiber membrane (with a fiber diameter less than 2 μm) was used as the porous substrate, which had the ideal properties mentioned above and an additional thermal stability. A mechanical CS/PVA blend selective layer was adopted to meet the separation demand.

## 2. Materials and Methods

### 2.1. Materials and Reagents

Poly (vinyl alcohol) being 87–89 mol% hydrolyzed and having a molecular weight of 84,000 (PVA 1788) was purchased from the Sinopec Group, China. The chitosan (with deacetylation ratios of H and LL at 84% and 85%, respectively) and γ-(2,3-epoxypropoxy) propyltrimethoxysilane were supplied by Aladdin Bio-Chem Technology Co., Ltd. (Shanghai, China) The ultra-fine glass fiber microporous membrane with a fiber diameter less than 2 μm and average pore diameters of 0.3 μm was purchased from Qingxin Ltd., Shenzhen China.

### 2.2. Membrane Preparation

PVA was dissolved in a 1 wt% acetic acid aqueous solution under vigorous stirring to dissolve completely. Then, a certain amount of chitosan was added into the PVA aqueous solution and dissolved for 4 h. The CS/PVA mixed solution was then cast using a casting knife with a gap of 30 μm onto the glass fiber microporous membrane substrate which had been immersion treated for 1 h with the 1 wt% ethanol solution of γ-(2,3-epoxypropoxy) propyltrimethoxysilane. The fabricating procedure for the CS/PVA-based composite membranes is illustrated in Figure 1. Finally, the composite membranes were dried at room temperature for at least 24 h to form a CS/PVA coating, and then dried in a vacuum oven at 80 °C for 24 h to remove the residual moisture. The CS/PVA mixed solution forming selective layers of the composite membranes had four different additions of PVA and CS such as 10 wt% PVA and 8 wt% PVA and 2 wt% CS, respectively; 6 wt% PVA and 4 wt% CS; and 4 wt% PVA and 6 wt% CS. The membranes derived from the four different CS/PVA additions were named CS0, CS2, CS4 and CS6, respectively. The effective thickness of the selective layer could be tuned by adjusting the amount of solution or the concentration of the CS/PVA. Part of the composite membrane CS4 was heat treated at 120 °C (named as CS4-120) or 180 °C (named as CS4-180) in an argon atmosphere to improve the separation performance, respectively.

### 2.3. Membrane Characterization

ATR-FTIR spectra were employed on a Nicolet iS5 FTIR spectrometer (Thermo Scientific, Waltham, MA. USA) over the wavenumber range of 400–4000 cm^−1^ to assess the characterization of the composite membranes. A thermo gravimetric analysis (TGA) was employed to assess the thermal stability of the PVA and chitosan from 50 °C to 900 °C in the argon atmosphere on a Netzch STA 449 F3 thermal analyzer, and the weight loss curves were determined. The surface morphology of the composite membranes was investigated by a Hitachi S-4800 field emission scanning electron microscope. The mechanical property of the composite membranes was tested at room temperature on a SHIMADIU AG-I 1KN at a strain rate of 5 mm·min^−1^, and the size of the membrane samples was 50 mm × 4 mm.

### 2.4. Gas Permeation Measurements

The gas permeance of the membranes for the pure gases of CO_2_ and N_2_ was conducted at room temperature based on the conventional constant volume/variable pressure technique [31]. The procedure was carried out on a permeation instrument constructed by Labthink, China. The general device setup of the gas permeation measurement system is shown in Appendix A. The gas flux was normalized by pressure to give the permeance by Equation (1):*P_i_/L = N_i_/Δp*(1)
where *P_i_/L* is the gas permeance with a unit of GPU (1 GPU = 10^−6^ cm^3^ (STP)/(cm^2^·s·cmHg)), *N_i_* is the steady-state flux of gas *i* through the membrane, *L* is the membrane thickness, and *Δp* is the transmembrane pressure difference [32]. The permeance (GPU) of either gas was obtained from the average value of at least 5 tests. The effective area of the test of membranes was 38.48 cm^2^.

The ideal selectivity (*α_ij_*) was calculated by Equation (2):(2)$$\alpha_{ij}=\frac{P_{i}}{P_{j}}$$

## 3. Results and Discussion

### 3.1. Characterization of Composite Membranes

The chemical structure of the composite CS/PVA-based membranes was characterized by ATR-FTIR and compared with the primary glass fiber microporous membrane substrate as shown in Figure 2. The CS/PVA-based composite membranes exhibited characteristic peaks at 660 cm^−1^ which was attributed to the crystalline sensitive band of chitosan [33]. The peaks at 1648 cm^−1^ and 1329 cm^−1^ were assigned to amide I and III bands in the chitosan, respectively, and the peak at 1560 cm^−1^ could be attributed to an amide II band in the chitosan [34,35]. The characteristic peak near 1251 cm^−1^ was attributed to the in-plane vibration of a secondary hydroxyl group in the PVA and chitosan chains [36]. The characteristic peaks at 1071 cm^−1^ and 1026 cm^−1^ were attributed to the stretch of the secondary hydroxyl groups in the PVA and chitosan segments, and the primary hydroxyl groups in the chitosan segments, respectively. Compared with the characteristic peaks of the free secondary hydroxyl group (1100 cm^−1^) and primary hydroxyl group (1050 cm^−1^), these two peaks of hydroxyl groups were at a lower frequency, attributed to the hydrogen bond between the PVA and chitosan segments. The ATR-FTIR spectra indicated that the chemical structure and functional group of the polymer coating had no obvious change after a heat treatment under 180 °C.

The weight loss curves of pure PVA and CS were determined by TGA as shown in Figure 3. Both the TGA curves showed a significant thermal stability under 230 °C. The two polymers both showed a major thermal loss step which indicated the decomposition of polymer main chains. The TGA indicated that the PVA and CS could withstand the heat treatment under 180 °C without significant decomposition.

The surface morphology of the membranes is shown in Figure 4. As shown in Figure 4A, the primary glass fiber microporous membrane with micron sized holes consisted of uniform and smooth glass fibers. After the surface treatment with a silane coupling agent, the glass fiber showed a good compatibility with the polymer coating. As shown in Figure 4C,D, a dense CS/PVA selective layer could be seen on the glass fiber microporous substrate.

Strain-stress curves of the CS/PVA composite membranes are shown in Figure 5. For the composite membranes, the tensile strength was higher than 3 MPa. The CS/PVA coating rendered the originally brittle glass fiber membrane with a good flexibility as shown in the inset of Figure 5. The strain-stress curves suggested that the CS/PVA composite membranes were strong enough for the gas separation process.

### 3.2. Effect of CS Content and Moisture on the Separation Performance of the Composite Membrane

The experimental results of the gas separation performance are shown in Figure 6. They indicate the effect of the CS content on the permeance and selectivity of the composite membranes under a 1 bar pressure difference. The permeance of the CO_2_ and N_2_ both increased with the increasing CS content which could be attributed to the decreasing crystallinity of the PVA in the polymeric coating caused by the addition of CS segments [22]. According to the reported literature, the change in the crystallinity of PVA membranes caused by blending with other polymers has been studied and it was found that there was an inverse relationship between the PVA crystallinity and the permeance of gases in PVA-blend membranes. The compact structure of polymer crystallites could hinder the sorption and diffusion of the permeant gas molecules in membranes compared with the amorphous phase [21]. The result of the gas separation measurement in our study was in accord with the situation above, namely, that the PVA crystallinity would go through a minimum at the content with the maximum gas permeance.

Following the facilitated transport mechanism, an increasing CO_2_ permeance occurs from the reversible reaction between the CO_2_ molecules and amine groups on CS segments acting as facilitated transport carriers [26]. The polymer membranes based on the facilitated transport mechanism are preferred due to the higher permeance and selectivity of CO_2_ rather than in the membranes only following the solution-diffusion mechanism. In the dry membranes, the CO_2_ permeance increased with the CS content, and reached a maximum 233 GPU at a 4 wt% addition of CS. In the dry composite membranes, the highest selectivity of the CO_2_/N_2_ reached 7.54 and largely exceeded that of the composite membrane with a pure PVA coating. The selectivity initially increased with the CS content and reached the maximum at 2 wt%, and then decreased with the further increase in CS.

To improve the separation performance, the composite membrane CS4 was suspended above an 80 °C water bath to make it moist. In the wetted membrane, the CO_2_ transport was improved compared to that in the dry membrane, while the N_2_ permeance decreased slightly compared to that in the dry membrane, resulting in a significant improvement of the CO_2_/N_2_ selectivity in wet environments. This result could be attributed to the increasing moisture content in the CS/PVA coating of the composite membrane, so that the CO_2_ could reversibly react with water to produce bicarbonate, facilitating the transport of CO_2_ in the membrane. Compared with N_2_, CO_2_ permeance increases significantly since its affinity with carrier groups is higher [37,38], making CO_2_/N_2_ more selective in a wetted membrane than in a dry membrane. Here, in the wetted membranes, the CO_2_ permeance increased with the CS content and reached a maximum 454 GPU at a 4 wt% addition of CS, while the highest selectivity of CO_2_/N_2_ reached 17.71 and largely exceeded that of the dry composite membranes. The selectivity initially increased with the CS content, reached a maximum at 4 wt%, and then decreased with a further increase in CS.

### 3.3. Effect of Operation Pressure on the Separation Performance of the Composite Membrane

The composite membrane CS4 had the best overall gas performance selectivity. This composite membrane was further tested under a different pressure difference as shown in Figure 7. It could be observed that the N_2_ permeance of the composite membrane did not change appreciably at an operation pressure higher than a 2 bar level, while the CO_2_ permeance and CO_2_/N_2_ selectivity decreased slightly with an increasing operation pressure, which may have been caused by the promotion of transmission mechanisms [37,39]. This phenomenon occurred in the separation process of the membrane based on the solution-diffusion model. The calculation of the permeance requires a division of the pressure difference between the feed side and permeate side. Under a small pressure difference, the effect of promoting the transmission mechanism is significant, and the CO_2_ carriers gradually reach saturation when the pressure difference increases. At this point, CO_2_ flux will gradually tend to stabilize the constant at a high pressure difference [39]. For the above reasons, the CO_2_ permeance and the pressure difference here showed a negative correlation as a whole. The highest CO_2_ permeance and CO_2_/N_2_ selectivity of the composite membrane CS4 reached 265 GPU and 8.27 at the 2 bar level.

### 3.4. Effect of Heat Treatment on the Separation Performance of the Composite Membrane

As shown in Figure 8, the heat-treated composite membrane CS4 at 120 °C and 180 °C showed an improved separation performance. The heat treatment would cause certain physical crosslinking by the rearrangement of polymer crystallites in the CS/PVA coating [22]. Consequently, the CO_2_ permeance of these heat-treated composite membranes decreased to 198 GPU and 187 GPU while the CO_2_/N_2_ selectivity increased, but then again, the chemical structure and functional group of the polymer coating had no obvious change after the heat treatment, as shown in Figure 2. Due to the presence of a large number of amino groups and hydroxyl groups in the CS/PVA composite coating, the heat-treated composite membrane showed a high CO_2_ permeance and selectivity for CO_2_/N_2_ while maintaining a good mechanical property [22]. On the other hand, the hydrogen bonds between the PVA and chitosan segments indicated by the results of the ATR-FTIR also maintained a high CO_2_/N_2_ selectivity.

## 4. Conclusions

In this study, a chitosan (CS) and PVA aqueous solution was coated to the surface of a glass fiber microporous membrane, thereby improving the CO_2_ permeance and CO_2_/N_2_ selectivity. The CS/PVA coating rendered the originally brittle glass fiber membrane with a good mechanical property and flexibility. The addition of CS to the coating increased the CO_2_ permeance. After a wetting treatment, the CO_2_ permeance and gas selectivity were higher than that in dry membranes because moisture promotes composite membrane transmission. The composite membrane CS4 was the best selection in this study for CO_2_ separation under a 1 bar pressure difference and humid environment, with the CO_2_/N_2_ selectivity reaching 17.71. After the heat treatment, the permeance of N_2_ decreased more significantly than that of the CO_2_, which led to an increase in the CO_2_/N_2_ selectivity. This method is based on environmentally sustainable materials and has a simple preparation process, representing an attractive alternative to traditional CO_2_ removal technologies with many practical applications.

## Figures and Tables

**Figure 1 membranes-13-00036-f001:**
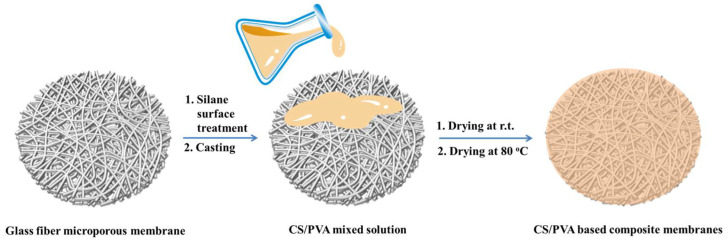
Illustration of the fabricating procedure for CS/PVA-based composite membranes.

**Figure 2 membranes-13-00036-f002:**
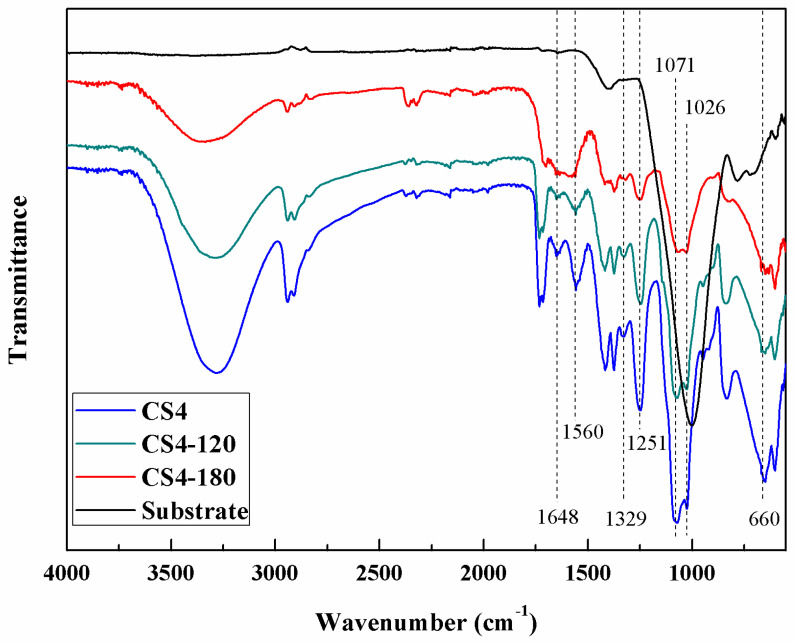
ATR–FTIR spectra of composite membranes.

**Figure 3 membranes-13-00036-f003:**
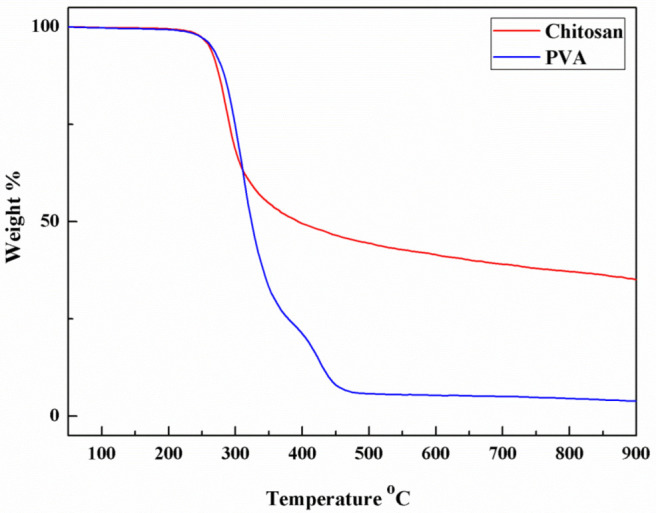
TGA curves of chitosan and PVA.

**Figure 4 membranes-13-00036-f004:**
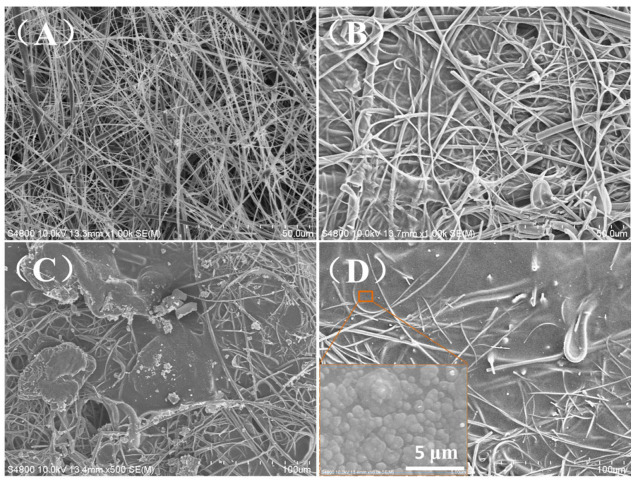
The surface morphology of the membranes. (**A**) glass fiber microporous membrane, (**B**) CS0, (**C**) CS4, and (**D**) CS6.

**Figure 5 membranes-13-00036-f005:**
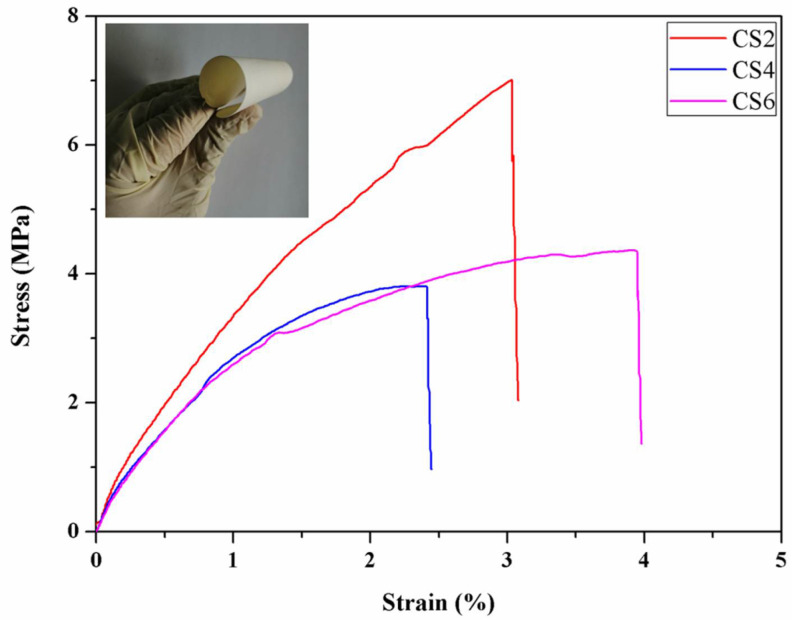
Strain-stress curves of the composite membranes.

**Figure 6 membranes-13-00036-f006:**
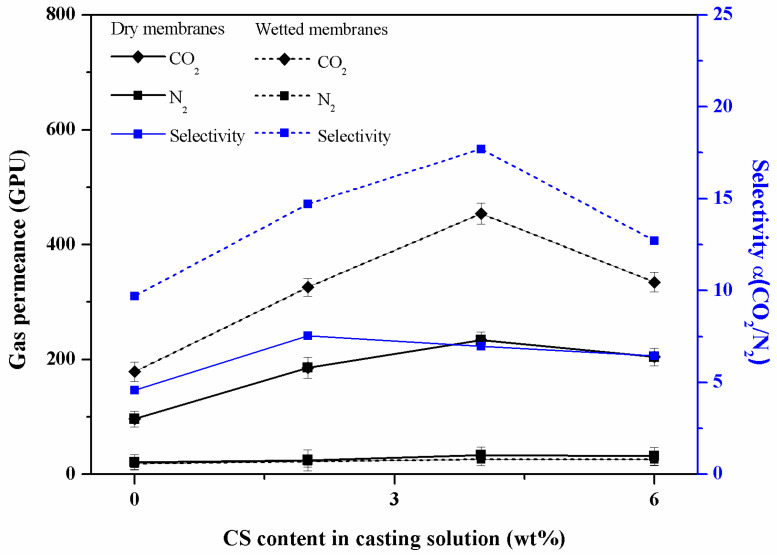
Effect of CS content and moisture on the separation performance of the composite membrane.

**Figure 7 membranes-13-00036-f007:**
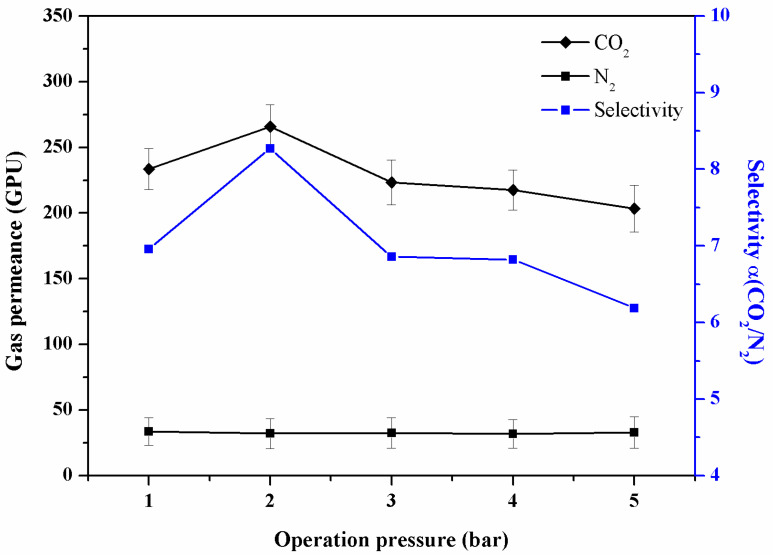
The relationship between gas permeance/selectivity and the operation pressure of the composite membrane CS4.

**Figure 8 membranes-13-00036-f008:**
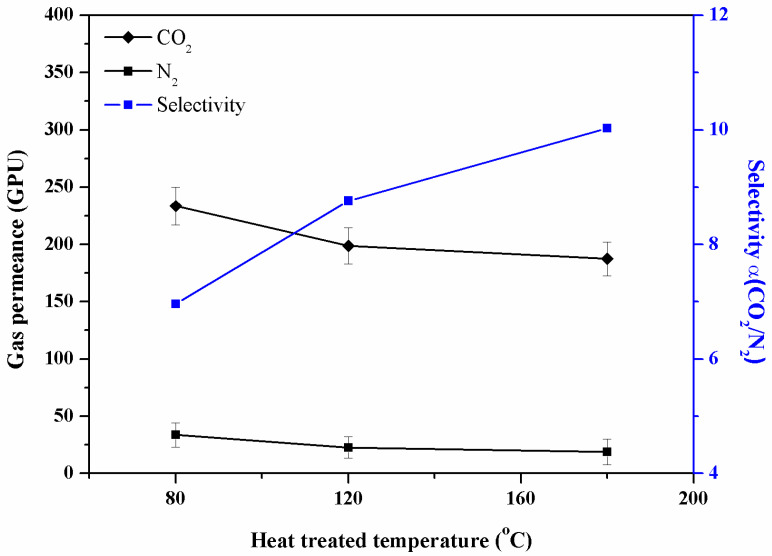
The gas permeance/selectivity of heat-treated composite membrane CS4 at 1 bar.

## Data Availability

Not applicable.

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
