# Peer review of "Preparation of a PVA/Chitosan/Glass Fiber Composite Membrane and the Performance in CO_2_ Separation"

_membranes, 2022, doi:10.3390/membranes13010036_

Round 1
Reviewer 1 Report
The authors have synthesized chitosan/PVA membranes on a glass fiber support. This work is certainly within the scope of Membranes and has some merit to being published, but I have several comments for the authors to consider.
1) The overall manuscript requires significant English revision. Although the topic of discussion appears credible, the meaning is sometimes difficult to interpret. Some sections are confusing, and sometimes I cannot understand what the authors want to say.
2) The introduction is adequate in scope, but I suggest the authors add more specific information. For example, Line 64-65 states polymer membranes have low separation factor, low permeability, and low thermal stability, but it is unclear to the reader what low means. Similarly, Line 85 states Cai et al. produced membranes with excellent selectivity, but again the reader does not know what excellent means. Is a selectivity of 10 excellent or 100? Line 87 states “ultra-thin” but is this 10 nm or 10 um for composite membranes? I suggest a thorough revision of the introduction with more specific examples so a reader can understand the relative performance of the various membranes discussed.
3) Line 115: What does TFC mean?
4) Why was chitosan/PVA chosen as the combination material? The authors should cite some of the other groups that have done work on this material. Has it ever been reported for CO2 (or any gas) separation membranes? It does have use in the medical industry, it seems, so the authors could reference this.
5) Line 137: Is propytrimethoxysilane missing a letter in “propyl?”
6) Line 146: “microporous membrane substrate which had been treated by the 1 wt% ethanol solution of γ-(2,3-epoxypropoxy) propytrimethoxysilane.” What treatment was performed? How are the acetic acid and ethanol solutions different?
7) Line 149: What do the authors mean by “the membranes were dried at room temperature and in a cavuum oven at 80 C for 24 h, respectively?” Was the sample dried once in air and then in a vacuum furnace or were different samples dried differently? Does 24 h apply to both drying processes? Rewrite to clarify the process.
8) Figure 1 is not helpful. It does not represent the casting step and instead shows a solution being poured over the gas fiber. It is too simple and not descriptive of the actual process. It should be more detailed with the actual steps involved, including the ethanol solution treatment, a depiction of the casting step and clarification on the drying process (temperature and gas atmosphere).
9) How was the solution changed to obtain the effective thickness? Did the authors change the concentration from the concentrations listed? It is unclear. A table summarizing the different membranes with synthesis differences would be helpful.
10) How were these membranes mounted for permeation testing? Can the authors include a description or a previous reference with details?
11) While common in polymer membrane research, the constant volume/variable pressure technique is not common in other membrane fields. A reference that explains the process is suggested to help unfamiliar readers looking for additional explanation.
12) Figure 2 (FTIR discussion): The authors have described several peaks but I am not convinced of their final statement “no obvious change under 180 C).” It is evident that the peak intensities change between samples and consistently reduce with increasing temperature. There also appears to be a slight shift in some peaks. The authors should label the figure with relevant peaks and more thoroughly discuss any differences between the samples and why peak intensities may vary. The authors should preferably use intensity ratios of each peak to show there is no significant change between samples.
13) The TGA curves for pure components were done in argon. Were the membranes also heated in argon? Clarify in the experimental section. It seems that operation would require oxygen, so would it not be better to run TGA in air as well? What about testing the membrane samples?
14) Line 216 states “after surface treatment with silane coupling agent, polymer coating and glass fiber show good compatibility.” What silane treatment was performed? This was not described previously.
15) Figure 5: What is the reason for CS2 to have higher stress and an intermediate strain breaking point compared to CS4 and CS6? Why was CS0 not tested?
16) If the addition of CS causes a change in crystallinity, could the authors use XRD to characterize the crystallinity and prove that 4% CS addition contains the least crystallinity?
17) The explanation on transmission mechanism is not clear. Can the authors define transmission mechanism in the text? If the CO2 carriers become saturated then the flux should become stable and the permeance should decline linearly with pressure. Is this observed? Perhaps the authors should show flux in these equations with permeance to explain the phenomenon.
18) The heat treatment may also affect crystallinity according to the authors’ arguments. The authors state FTIR did not show a difference but the peak intensities changed. Could the increase in performance be attributed to less crystallinity? Again, could the authors perform a measurement on the crystallinity directly, such as XRD?
19) Figure 8 right axis label contains a mistake in letters. Bottom axis needs unit in parentheses.
20) The authors never mention the membrane thickness. Is it possible the thickness changed between samples, which could affect the measured permeance? Why did the authors not measure the thickness with SEM or another technique?
Author Response
Dear Reviewer:
Thank you for your comments concerning our manuscript entitled “Preparation of PVA/chitosan/glass fiber composite membrane and the performance in CO2 separation” (No.: membranes-2118324). These comments are all valuable and very helpful for revising and improving our paper, as well as the important guiding significance to our researches. We have studied comments carefully and have made correction which we hope meet with approval. Revised portion is marked up using the “Track Changes” function in the revised manuscript. Special thanks to you for your positive comments. The main corrections in the paper and the responds to your comments are as following:
1) The overall manuscript requires significant English revision. Although the topic of discussion appears credible, the meaning is sometimes difficult to interpret. Some sections are confusing, and sometimes I cannot understand what the authors want to say.
Response: We are very sorry for the inaccurate statement in the manuscript and have made serious revisions.
2) The introduction is adequate in scope, but I suggest the authors add more specific information. For example, Line 64-65 states polymer membranes have low separation factor, low permeability, and low thermal stability, but it is unclear to the reader what low means. Similarly, Line 85 states Cai et al. produced membranes with excellent selectivity, but again the reader does not know what excellent means. Is a selectivity of 10 excellent or 100? Line 87 states “ultra-thin” but is this 10 nm or 10 um for composite membranes? I suggest a thorough revision of the introduction with more specific examples so a reader can understand the relative performance of the various membranes discussed.
Response: Thanks for your suggestion. We add more specific information and data in the introduction.
3) Line 115: What does TFC mean?
Response: I am sorry for this mistake in expression. We have revised the statement here.
4) Why was chitosan/PVA chosen as the combination material? The authors should cite some of the other groups that have done work on this material. Has it ever been reported for CO2 (or any gas) separation membranes? It does have use in the medical industry, it seems, so the authors could reference this.
Response: Because both chitosan and PVA are commonly used and water-soluble materials in gas separation membrane, we chose them as the combination materials. And during the experiment, chitosan/PVA showed good compatibility in water and good film forming property. But there are few references about application of blending chitosan/PVA in gas separation.
5) Line 137: Is propytrimethoxysilane missing a letter in “propyl?”
Response: We are very sorry for this mistake. We have corrected it in the revised manuscript.
6) Line 146: “microporous membrane substrate which had been treated by the 1 wt% ethanol solution of γ-(2,3-epoxypropoxy) propytrimethoxysilane.” What treatment was performed? How are the acetic acid and ethanol solutions different?
Response: The treatment mentioned here is “immersing treated”, and we described this in detail in the revised manuscript.
The acetic acid was used to prepare CS/PVA solution for coating layer, the ethanol solution of γ-(2,3-epoxypropoxy) propytrimethoxysilane was just used in the treatment of glass fiber microporous membrane substrate to improve its compatibility with CS/PVA.
7) Line 149: What do the authors mean by “the membranes were dried at room temperature and in a cavuum oven at 80 C for 24 h, respectively?” Was the sample dried once in air and then in a vacuum furnace or were different samples dried differently? Does 24 h apply to both drying processes? Rewrite to clarify the process.
Response: Thanks for your suggestion. We rewrite this process in the revised manuscript.
8) Figure 1 is not helpful. It does not represent the casting step and instead shows a solution being poured over the gas fiber. It is too simple and not descriptive of the actual process. It should be more detailed with the actual steps involved, including the ethanol solution treatment, a depiction of the casting step and clarification on the drying process (temperature and gas atmosphere).
Response: Thanks for your suggestion. We redraw Figure 1 in more detail in the revised manuscript.
9) How was the solution changed to obtain the effective thickness? Did the authors change the concentration from the concentrations listed? It is unclear. A table summarizing the different membranes with synthesis differences would be helpful.
Response: In the SEM images of the membrane cross section, there was no obvious and explicit skin layer was observed, therefore, thickness of the membrane was not discussed in the manuscript. This may be due to the large pore diameter of the glass fiber membrane substrate that much larger than the surface pore of the common Psf porous membrane substrate.
Therefore, we did not consider the thickness changed between samples, but controlled the film forming conditions strictly to eliminate thickness errors between samples.
10) How were these membranes mounted for permeation testing? Can the authors include a description or a previous reference with details?
Response: We supplement references about permeation testing in the secion 2.4 of the revised manuscript.
A general apparatus for the gas permeation measurement via constant volume/variable pressure method consists of a membrane test cell with defined permeate test volume and a vacuum pump, which is connected through a valve to the test volume. For the pressure detection of the feed side and permeate side, two pressure sensors are additionally needed. The measurement begins with the evacuation of the test volume by the vacuum pump. After the evacuation, the test gas which permeates through the membrane induces a slight pressure increase with time in the test volume on the permeate side.
11) While common in polymer membrane research, the constant volume/variable pressure technique is not common in other membrane fields. A reference that explains the process is suggested to help unfamiliar readers looking for additional explanation.
Response: Thanks for your suggestion. This is really necessary, and we supplement references about permeation testing in the secion 2.4 of the revised manuscript.
12) Figure 2 (FTIR discussion): The authors have described several peaks but I am not convinced of their final statement “no obvious change under 180 C).” It is evident that the peak intensities change between samples and consistently reduce with increasing temperature. There also appears to be a slight shift in some peaks. The authors should label the figure with relevant peaks and more thoroughly discuss any differences between the samples and why peak intensities may vary. The authors should preferably use intensity ratios of each peak to show there is no significant change between samples.
Response: Thanks for your suggestion and we label the figure 2 with characteristic peaks.
We noticed that the three originally separated peaks between 1750 cm-1 and 1500 cm-1 revealed a broad peak in the range of 1750 cm-1-1500 cm-1, after heating treatment at 180oC. This broad peak is difficult to discuss the attribution to clear groups and chemical structure change. On the other hand, other peaks could be attributed to corresponding main groups of CS and PVA, and they didn't show significant change in position.
As for the intensity of the FTIR peaks, we don't think it could be used to discuss changes in chemical structure without convincing normalization to the FTIR spectra. So we didn't discuss the change of peak intensity.
13) The TGA curves for pure components were done in argon. Were the membranes also heated in argon? Clarify in the experimental section. It seems that operation would require oxygen, so would it not be better to run TGA in air as well? What about testing the membrane samples?
Response: The composit membranes were heated in argon as well. And we clarified this operation condition in the revised manuscript. The separation test of the membrane samples was at room temperature.
If the heat treatment was conducted in oxygen or air, the separation performance and structure might be impact. In order to avoid too complex factors affecting the experimental system, we try to avoid the influence of oxygen in this study.
14) Line 216 states “after surface treatment with silane coupling agent, polymer coating and glass fiber show good compatibility.” What silane treatment was performed? This was not described previously.
Response: We describe the surface treatment more detailed in section 2.2 of the revised manuscript, similar to the answer to Question 6.
15) Figure 5: What is the reason for CS2 to have higher stress and an intermediate strain breaking point compared to CS4 and CS6? Why was CS0 not tested?
Response: We think it may be attributed to decreased crystallinity of PVA membranes caused by blending with CS. This study is mainly about the separation effect of CS/PVA coating, therefore, CS0 (with pure PVA coating) was not tested.
16) If the addition of CS causes a change in crystallinity, could the authors use XRD to characterize the crystallinity and prove that 4% CS addition contains the least crystallinity?
Response: I am sorry for that our laboratory is temporarily unavailable due to the COVID-19, the campus is closed. Within the time limit of manuscript revision, this situation is difficult to change.
17) The explanation on transmission mechanism is not clear. Can the authors define transmission mechanism in the text? If the CO2 carriers become saturated then the flux should become stable and the permeance should decline linearly with pressure. Is this observed? Perhaps the authors should show flux in these equations with permeance to explain the phenomenon.
Response: As shown in Figure 7, the permeance showed approximately linear descent with pressure (2 bar- 5 bar).
18) The heat treatment may also affect crystallinity according to the authors’ arguments. The authors state FTIR did not show a difference but the peak intensities changed. Could the increase in performance be attributed to less crystallinity? Again, could the authors perform a measurement on the crystallinity directly, such as XRD?
Response: As mentioned in the answer to question 12, we don't think the intensity of the FTIR peaks could be used to discuss changes in chemical structure without convincing normalization to the FTIR spectra. So we didn't discuss the change of peak intensity attributed to the crystalline structure.
About XRD, as mentioned in the answer to question 16, our laboratory is temporarily unavailable due to the COVID-19, the campus is closed. Within the time limit of manuscript revision, this situation is difficult to change.
19) Figure 8 right axis label contains a mistake in letters. Bottom axis needs unit in parentheses.
Response: We are sorry for the mistakes and correct in the revised manuscript.
20) The authors never mention the membrane thickness. Is it possible the thickness changed between samples, which could affect the measured permeance? Why did the authors not measure the thickness with SEM or another technique?
Response: In the SEM images of the membrane cross section, there was no obvious and explicit skin layer was observed, therefore, SEM images of the membrane cross section were not shown in the manuscript. This may be due to the large pore diameter of the glass fiber membrane substrate that much larger than the surface pore of the common PSf porous membrane substrate.
Therefore, we did not consider the thickness changed between samples, but controlled the film forming conditions strictly to eliminate errors between samples.
Reviewer 2 Report
Overall:
Both the permeance and selectivity of these membranes are too low to be of economic interest. Otherwise the paper is an acceptable generic membrane material study.
To publish this work as a materials paper, the authors need to calculate permeability of the membrane material. This can be done by measuring the thickness of their membranes (to convert permeance to permeability) and weighing the glass support before and after membrane fabrication to calculate the volume fraction of their membranes that are the polymeric material.
The materials studied and properties measured (CO2/N2 selectivity <10) have low interest to the field. I suspect it would not be of interest to study these materials for other gas pairs, nor make further variations of these materials - so I do not recommend further work be carried out.
According to the publication criteria of this journal, I recommend acceptance after major revisions. Specifically, improved description of the membrane testing procedure and proof of calibration against a standard/known material.
Abstract:
The abstract needs quantitative information included. I.e.
Membrane permeability and selectivity; before and after heat treatment.
Introduction:
The English is passable. If Membranes has a service to improve the English of the manuscript, then I recommend this paper be passed through it.
For example Line 29: "The gas separation technology" could be "Membrane technology"
Line 117: The authors should remove claims of a economically viable membrane. There are no references to process designs to evaluate the economics or what membrane properties would be required to achieve an economically viable membrane.
Additionally, there is little relevance of the properties studied in this work to an economically viable membrane. For example, the casting method is unsuitable for creating thin films required for high permeance.
Experimental section:
The authors should provide a piping and instrumentation diagram of their Labthink instrument. Or provide further documentation or reference to an experimental procedure and data analysis method. This could be provided in a supporting information document.
The authors should provide documentation of how their membrane rig was calibrated against a membrane of known permeability properties.
The TGA curve in Figure 3 should include the data for the composite membrane materials produced (in addition to the precursors shown).
Author Response
Dear Reviewer:
On behalf of all the contributing authors, I would like to express our sincere appreciations of your constructive comments concerning our article entitled "Preparation of PVA/chitosan/glass fiber composite membrane and the performance in CO2 separation" (No.: membranes-2118324). These comments are all valuable and helpful for improving our article. According to these comments, we have made extensive modifications to our manuscript to make our results convincing. In this revised version, revised portion is marked up using the "Track Changes" function. Special thanks to you for your positive comments. The main corrections in the paper and the responds to your comments are as following:
Abstract:
The abstract needs quantitative information included. I.e. Membrane permeability and selectivity; before and after heat treatment.
Response: Thank you for your valuable advice. We add quantitative information in the abstract of the revised manuscript.
Introduction:
The English is passable. If Membranes has a service to improve the English of the manuscript, then I recommend this paper be passed through it.
For example Line 29: "The gas separation technology" could be "Membrane technology"
Line 117: The authors should remove claims of a economically viable membrane. There are no references to process designs to evaluate the economics or what membrane properties would be required to achieve an economically viable membrane.
Additionally, there is little relevance of the properties studied in this work to an economically viable membrane. For example, the casting method is unsuitable for creating thin films required for high permeance.
Response: Thank you for your positive comments. And we have deleted the inappropriate statement and revised the manuscript according to your opinion.
Experimental section:
The authors should provide a piping and instrumentation diagram of their Labthink instrument. Or provide further documentation or reference to an experimental procedure and data analysis method. This could be provided in a supporting information document.
The authors should provide documentation of how their membrane rig was calibrated against a membrane of known permeability properties.
The TGA curve in Figure 3 should include the data for the composite membrane materials produced (in addition to the precursors shown).
Response: Thank you for your valuable advice. And we will provide in a supporting information document.
Considering that glass fiber has good thermal stability and the low content of polymers in the the composite membrane, we did not test thermal stability of the composite membranes. I am sorry for that our laboratory is temporarily unavailable due to the COVID-19, the campus is closed. Within the time limit of manuscript revision, this situation is difficult to change. It is difficult to supplement TGA curve of the composite membrane.
Reviewer 3 Report
The article is presenting the preparation and characterization of the glass fiber microporous coated by Chitosan and PVA aqueous solution to improve CO2 permeance and CO2/N2 selectivity.
Specific comments are listed below:
1. Illustration of the experimental setup for the Gas permeation measurement should be presented.
2. Please add error bar for Fig. 6, 7 and 8.
3. Please add SEM characterization, especially for the cross sectional view of the membranes
4. Many references are outdated, not from recent five years.
Author Response
Dear Reviewer:
Thank you for your comments concerning our manuscript entitled “Preparation of PVA/chitosan/glass fiber composite membrane and the performance in CO2 separation” (No.: membranes-2118324). These comments are all valuable and very helpful for revising and improving our paper, as well as the important guiding significance to our researches. We have studied comments carefully and have made correction which we hope meet with approval. Revised portion is marked up using the “Track Changes” function in the revised manuscript. Special thanks to you for your positive comments. The main corrections in the paper and the responds to your comments are as following:
The article is presenting the preparation and characterization of the glass fiber microporous coated by Chitosan and PVA aqueous solution to improve CO2 permeance and CO2/N2 selectivity.
Specific comments are listed below:
- Illustration of the experimental setup for the Gas permeation measurement should be presented.
Response: Thanks for your suggestion. We supplemented illustration of the experimental setup and put it in the Supporting Information if we can get your approval.
- Please add error bar for Fig. 6, 7 and 8.
Response: Thanks for your suggestion. And error bar has been added for Fig. 6, 7 and 8 in the revised manuscript.
- Please add SEM characterization, especially for the cross sectional view of the membranes
Response: In the SEM images of the membrane cross section, there was no obvious and explicit skin layer was observed, therefore, thickness of the membrane was not discussed in the manuscript. This may be due to the large pore diameter of the glass fiber membrane substrate that much larger than the surface pore of the common Psf porous membrane substrate. We supplemented the SEM images of the membrane cross section and put it in the Supporting Information if we can get your approval.
- Many references are outdated, not from recent five years.
Response: Thanks for your suggestion. Most of outdated references have been updated in the revised manuscript.
Reviewer 4 Report
Abstract
“ATR-FTIR spectra show that the chemical structure and functional group of the polymer coating had no obvious change under 180 °C”
- Was ATR-FTIR used to characterize the composite in the temperature range? How?
“As heat treatment, nitrogen permeance decreases by more than CO2 19 permeance, which leads to an increase in CO2/N2 selectivity”
-This statement is unclear
Manuscript
“Over the past decade, membrane-based CO2 separation from the mixture with light gases, such as O2, N2 and CH4, is a widely valued technology of industrial and technological advantage owing to the characteristic of membrane separation technology, for instance, environmental sustainability, simple process, scale adaptability and high energy efficiency”
- This statement is too long
“For the pressure of the international carbon emission reduction”
“The result of our study was in accord with the situation above”
- These statements are unclear
The manuscript needs to be thoroughly edited.
Author Response
Dear Reviewer:
Thank you for your comments concerning our manuscript entitled “Preparation of PVA/chitosan/glass fiber composite membrane and the performance in CO2 separation” (No.: membranes-2118324). These comments are all valuable and very helpful for revising and improving our paper, as well as the important guiding significance to our researches. We have studied comments carefully and have made correction which we hope meet with approval. Revised portion is marked up using the “Track Changes” function in the revised manuscript. Special thanks to you for your positive comments. The main corrections in the paper and the responds to your comments are as following:
Abstract
“ATR-FTIR spectra show that the chemical structure and functional group of the polymer coating had no obvious change under 180 °C”
- Was ATR-FTIR used to characterize the composite in the temperature range? How?
Response: We are sorry for this inaccurate expression and correct it in the revised manuscript.
“As heat treatment, nitrogen permeance decreases by more than CO2 19 permeance, which leads to an increase in CO2/N2 selectivity”
-This statement is unclear
Response: We rewrite this statement in the revised manuscript.
Manuscript
“Over the past decade, membrane-based CO2 separation from the mixture with light gases, such as O2, N2 and CH4, is a widely valued technology of industrial and technological advantage owing to the characteristic of membrane separation technology, for instance, environmental sustainability, simple process, scale adaptability and high energy efficiency”
- This statement is too long
Response: Thank you for the advice. We rewrite this statement in the revised manuscript.
“For the pressure of the international carbon emission reduction”
“The result of our study was in accord with the situation above”
- These statements are unclear
Response: Thank you for the advice. We rewrite these statements in the revised manuscript.
The manuscript needs to be thoroughly edited.
Response: We are very sorry for the inaccurate statement in the manuscript and have made serious revisions.
Round 2
Reviewer 1 Report
I thank the reviewers for their efforts to answer my concerns and improve their manuscript. Please find some points below concerning the responses.
1. The English still has many mistakes, but it may be passable after MDPI performs an editing service, which I believe occurs after acceptance. A question for the authors: do you use Microsoft Word or some other software for the editing? Word has built-in grammar and spelling correction that would certainly notify you of many mistakes in this manuscript. You may need to load an English dictionary and change the settings to correct for English, but I think the authors should invest some time to using such a feature. There are also online tools for this task that the authors may be able to use. Some mistakes may be missed, but many mistakes would be corrected.
2. I notice the right axis of Fig. 6 has an "a" instead of "alpha" symbol. This was previously a mistake in Fig. 8 but the authors corrected it. Please carefully review your figures from whatever software you use because it seems you are making the same mistake repeatedly.
3. The authors have included the GPU of membranes in the abstract, which is a nice addition. However, I would suggest the authors consider their significant digits in reported measurements. Do the authors think there is a difference in 233.45 and 233.46 GPU? Is their experimental system capable of +/- 0.01 GPU measurement precision? I would guess that they can only reliable reproduce measurements around 233 +/- 1 GPU so there is no reason to report the .45. All measurements should be reported with consideration to the known accuracy.
4. It is sad that China continues its lockdowns and this is an excuse for not fully addressing some reviewer comments. While I understand, I do not like it as an answer. It would be better for the authors to look for literature examples that can explain results if they cannot run the experiments themselves. However, I understand the situation and I will only ask that my above comments be considered for acceptance.
Author Response
Thank you for your kind letter and constructive comments concerning our article (Manuscript No.: membranes-2118324).These comments are valuable and helpful for improving our article. All the authors have seriously discussed about these comments. According to the comments, we have tried best to modify our manuscript to meet with the requirements of journal. Revised portion is marked up using the “Track Changes” function in the revised manuscript.
- The English still has many mistakes, but it may be passable after MDPI performs an editing service, which I believe occurs after acceptance. A question for the authors: do you use Microsoft Word or some other software for the editing? Word has built-in grammar and spelling correction that would certainly notify you of many mistakes in this manuscript. You may need to load an English dictionary and change the settings to correct for English, but I think the authors should invest some time to using such a feature. There are also online tools for this task that the authors may be able to use. Some mistakes may be missed, but many mistakes would be corrected.
Response: Thanks for your suggestions. We feel sorry for our poor writings, and we do invite friends of us to help polish our article. Due to our friend’s help, the article was edited extensively. And we hope the revised manuscript could be acceptable for you.
- I notice the right axis of Fig. 6 has an "a" instead of "alpha" symbol. This was previously a mistake in Fig. 8 but the authors corrected it. Please carefully review your figures from whatever software you use because it seems you are making the same mistake repeatedly.
Response: We are sorry for these mistakes. We have carefully reviewed and corrected the manuscript.
- The authors have included the GPU of membranes in the abstract, which is a nice addition. However, I would suggest the authors consider their significant digits in reported measurements. Do the authors think there is a difference in 233.45 and 233.46 GPU? Is their experimental system capable of +/- 0.01 GPU measurement precision? I would guess that they can only reliable reproduce measurements around 233 +/- 1 GPU so there is no reason to report the .45. All measurements should be reported with consideration to the known accuracy.
Response: Thank you for your professional advice. And we have revised the data of permeance in the manuscript.
- It is sad that China continues its lockdowns and this is an excuse for not fully addressing some reviewer comments. While I understand, I do not like it as an answer. It would be better for the authors to look for literature examples that can explain results if they cannot run the experiments themselves. However, I understand the situation and I will only ask that my above comments be considered for acceptance.
Response: Once again, I sincerely apologize for this. According to current expectations, the situation could change two weeks later. Could the data be supplemented in the form of corrigendum, if we can get your permission.
Reviewer 2 Report
Congratulations on your publication.
Author Response
Thank you for your kind letter and for reviewers’ constructive comments concerning our article (Manuscript No.: membranes-2118324).These comments are all valuable and helpful for improving our article.
Reviewer 4 Report
Accept.
Author Response
We feel great thanks for your professional review work on our article. (Manuscript No.: membranes-2118324).These comments are all valuable and helpful for improving our article.